# Is the relative age effect just a European problem? A comprehensive analysis of birth date distribution and its impact on player selection at the 2023 FIFA Women's World Cup

Benito Pérez-González[1], Jairo León-Quismondo [2], José Bonal[2], Iyán Iván-Baragaño [2]*, Álvaro Fernández-Luna [2], Pablo Burillo [2,3]

**1** Faculty of Business and Communication, Universidad Internacional de La Rioja, Logroño, Spain, **2** Department of Sports Sciences, Faculty of Medicine, Health and Sports, Universidad Europea de Madrid, Madrid, Spain, **3** Department of Real Madrid Graduate School, Faculty of Medicine, Health and Sports, Universidad Europea de Madrid, Madrid, Spain

\* iyanivanbaragano@gmail.com

## Abstract

This study examines the presence and implications of the Relative Age Effect (RAE) in the 2023 FIFA Women's World Cup, focusing on the distribution of players' birth dates across continents and professional levels. Utilizing a sample of 735 players, a Poisson regression was conducted on the weekly birth frequency to determine the significance of the fit to a Poisson regression curve. Despite the lack of a significant RAE across the overall player population, our findings reveal notable geographical and positional variations. Specifically, European players, particularly goalkeepers and defenders, alongside the top 4 classified teams and nations with a substantial number of federation licenses, exhibit a significant RAE, suggesting a substantial impact on player selection favouring those born earlier in the year. Conversely, players from Africa, America, Asia, and Oceania do not show a significant RAE, indicating variability in its manifestation across different football environments.

## Introduction

The number of women and girls playing organized football has increased by 24% between 2019 and 2023 [1,2]. Similarly, the professionalization of many clubs and federations has led to an increase in the number of professional players to 19,064 [1].This fact, among other factors, has contributed to an improvement in the technical and tactical performance of many teams in the last decade [3]. Furthermore, this improvement in the performance of many teams has coincided with an increase in social and media interest. All these factors may signify a paradigm shift in the development and detection of talent in women's football. One of these factors could be the emergence of the Relative Age Effect [RAE]. All these factors may signify a paradigm shift in the development and detection of talent in women's football, highlighting the need to understand the predictors and mediating factors that influence athletic success.

**Data availability statement:** The data that support the findings of this study were obtained from LiveFutbol provider [https://www.livefutbol.com/competicion/frauen-wm] and is additionally available from the corresponding author [II-B], upon reasonable request.

**Funding:** The author(s) received no specific funding for this work.

**Competing interests:** The authors have declared that no competing interests exist.

The identification and development of talent in women's football has become an increasingly critical area of focus. This process involves understanding various predictors and mediating factors that influence athletic success. Physical attributes such as strength, speed, and endurance have been highlighted as essential predictors in studies [4], which explored physiological demands placed on elite female players and how these are linked to success at the highest levels of competition. Similarly, technical skills, including ball control, passing accuracy, and decision-making under pressure, have also been emphasized. For instance, Harkness-Armstrong et al. [5] examined the role of technical abilities in distinguishing between high-performing and lower-performing players. Despite these advances, research specific to talent identification in female football remains scarce [6,7] with gaps in literature focusing on youth female soccer and the factors influencing their progression to elite levels. Calls for more female-specific research [8,9], highlight the urgent need to address this disparity.

Since the appearance of the first publications on the RAE [10,11], this bias has been widely studied in numerous team sports such as football [12] or hockey [13], but also in individual sports like alpine skiing [14] or in athletic disciplines such as long jump [15]. In the case of team sports, there is evidence of the existence of a RAE on team performance, as well as in talent selection at early ages [16]. However, the number of studies conducted with male participants is clearly higher than those carried out with females [17]. For this reason, the influence of this variable in women's sports, and specifically in women's football, is not as evident.

In recent years, various studies have been conducted on female samples, although the results have been contradictory. One of the first studies carried out was published by Delorme et al. [18] in France. These authors analyzed the birth quarter of over 15,000 players registered with the French Football Federation, observing a generalized effect in favour of the first quarters of the year. However, this study showed that the effect was greater in lower categories compared to adults [19]. In Spain, this effect was also analyzed by conducting a cross-sectional analysis of over 4,000 players [20], finding that the relative age effect was more frequent as the level of competition increased. This study showed that 34.5% of the players called up by the national team were born in the first quarter of the year, compared to only 14% born in the last quarter, consistent with more recent findings in the Japanese League [21].

In contrast to previous studies, there have been investigations that have not found this effect in different female samples in big tournaments or in the Brazilian leagues [22–24]. The case of elite women's football is striking, as it is noteworthy that no relative age effect was observed in the FIFA U17 Women's World Cups in the editions of 2008, 2010, 2012, or 2014 [23], nor in the editions of the FIFA Women's World Cup held between 2007 and 2019. This study, which represents one of the investigations on this topic with an elite sample, aligned with other studies conducted with smaller samples where no such effect was observed [22,24].

In addition to these findings, recent research has further expanded the understanding of the RAE in female football, particularly at the international level. Andrew et al. [25] analyzed U17, U19, and senior women's soccer players participating in European Championships, revealing that RAE persisted in male but not female soccer players. Later research by Finnegan et al. [26] examined youth national team players in the United States and highlighted how early talent identification processes often tend to prefer players born earlier in the selection year. Similar patterns were reported by Morgans et al. [27] in their study of youth and senior players in Wales, where the RAE influenced not only team composition but also playing position. Also, Götze and Hoppe [28] explored the impact of RAE on female soccer players in Germany, confirming a consistent bias across competitive levels.

Therefore, it can be considered that the knowledge about the relative age effect in women's football, and elite football in particular, is still limited. For this reason, the objective of this study was to investigate the presence of the Relative Age Effect [RAE] in Women's Football

World Cup 2023 participants. Specifically, the aim was to examine the distribution of players' birth dates across continents and analyzing the influence of RAE by demarcation.

## Materials and methods

### Design and sample

The sample (n = 735) is represented by players from the Women's Football World Cup 2023. We obtained the information of players on the specialized platform Livefutbol [https://www.livefutbol.com/competicion/frauen-wm/]. The cutoff date is 1 January for all players, except for UK and Australian players whose cutoff date is 1 September.

### Data analysis

The Relative Age Effect (RAE) was detected through Poisson regression [29]. The Poisson regression formula y = e (b0 + b1x) serves to explain the frequency count of an event (y) by an explanatory variable x. The data used for Poisson regression were week of birth (WB) whereby the first week in January was designated WB 1, and time period of birth (Tb) describing how far from the beginning of the year a player was born. This last index ranging between 0 and 1 was calculated as Tb = (WB − 0.5)/52. In the Poisson regression, the event (y) was the frequency of birth in a given week and the explanatory variable (x) was Tb. We also calculated the index of discrimination (ID) according to Doyle and Bottomley [29] as e-b1. This index measures the relative odds of a player born on day 1 versus day 365 of the competition year being selected. The likelihood ratio D2 was determined according to Cohen et al. [30]. All statistical tests, including descriptive analysis, were performed using the software package R (version 4.3.2). Significance was set at $p < 0.05$.

## Results

The results suggest a notable distribution of players' birth dates according to their quartile (Q) or semester (Se) of birth by continent and classification, as shown in Table 1.

**Table 1. Birth date distributions according to their quartile (Q) or semester (Se) of birth by continent and classification.**

| | | Q1 | Q2 | Q3 | Q4 | Se 1 | Se 2 |
|---|---|---|---|---|---|---|---|
| Overall Players (n = 735) | n | 192 | 195 | 164 | 184 | 387 | 348 |
| | % | 26 | 27 | 22 | 25 | 53 | 47 |
| African Players (n = 297) | n | 22 | 28 | 20 | 22 | 50 | 42 |
| | % | 24 | 30 | 22 | 24 | 54 | 46 |
| American Players (n = 207) | n | 52 | 46 | 51 | 58 | 98 | 109 |
| | % | 25 | 22 | 25 | 28 | 47 | 53 |
| Asian Players (n = 115) | n | 33 | 27 | 26 | 29 | 60 | 55 |
| | % | 29 | 23 | 23 | 25 | 52 | 48 |
| European Players (n = 275) | n | 76 | 76 | 58 | 65 | 152 | 123 |
| | % | 28 | 28 | 21 | 23 | 56 | 44 |
| Oceanian Players (n = 46) | n | 9 | 18 | 9 | 10 | 27 | 19 |
| | % | 20 | 39 | 20 | 21 | 59 | 41 |
| Top 4 (n = 92) | n | 25 | 30 | 18 | 19 | 55 | 37 |
| | % | 27 | 33 | 20 | 20 | 60 | 40 |
| Federation Licenses > 100,000 (n = 229) | n | 65 | 62 | 51 | 51 | 80 | 58 |
| | % | 28 | 28 | 22 | 22 | 56 | 44 |

The data reveal a predominant presence of players in the first semester across the overall player sample. Specifically, African players exhibit a stronger presence in the first semester across the entire sample, with three out of four countries showing a majority of first-semester players and one country displaying more second-semester players.

Asian, European, and Oceanian players similarly show a predominance of first-semester births in the overall sample, with the majority of countries in these continents following this trend. In European countries, 9 of the 12 have more first-semester players, and 3 of them more second semester players. In Asian countries, 4 of the 5 nations have more first-semester players, and 1 of them more second semester players. Similarly, in Oceania, the two countries represented exhibit a higher prevalence of first-semester births.

Conversely, American players present a higher frequency of second-semester players in the entire sample. By countries, 4 of the 9 have more first-semester players, and 5 of them more second semester players. Nonetheless, the two countries with the most licenses, the United States and Canada, have more first-semester players.

Furthermore, the top four countries classified in the Women's Football World Cup 2023 exhibited a higher number of players born in the first semester. Additionally, countries with more than 100,000 Federation Licenses also presented a greater presence of first-semester players.

Fig 1 illustrates the quarter of birth distribution for the overall players participating in the Women's Football World Cup 2023. Fig 2 displays the frequency of week of birth (WB) for all players, European players, the top 4 classified, and countries with more than 100,000 Federation Licenses, respectively, accompanied by Poisson Regression analyses.

Table 2 presents the Poisson regression analysis of the Relative Age Effect (RAE) by frequency for all players in the Women's Football World Cup 2023.

Our analysis did not identify a significant presence of RAE across all players ($p = 0.11$). However, significant RAEs were observed in European players, the top 4 classified, and countries with more than 100,000 Federation Licenses in women's football ($p < 0.05$), indicating a notable impact of birth date on player selection in these groups. Conversely, players from Africa, America, Asia, and Oceania did not exhibit a significant RAE.

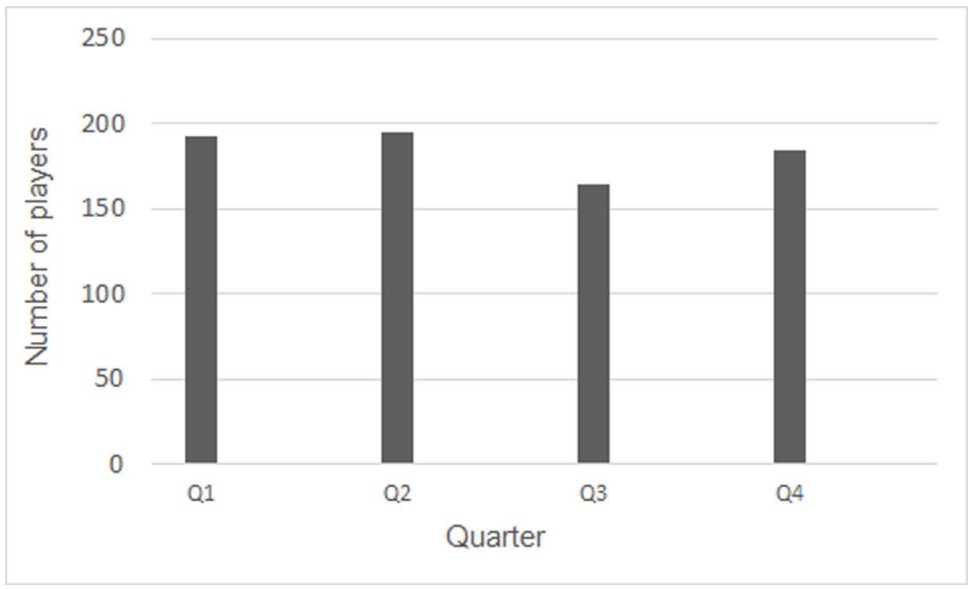

**Fig 1. Quarter of birth of overall players of Women's Football World Cup 2023.**

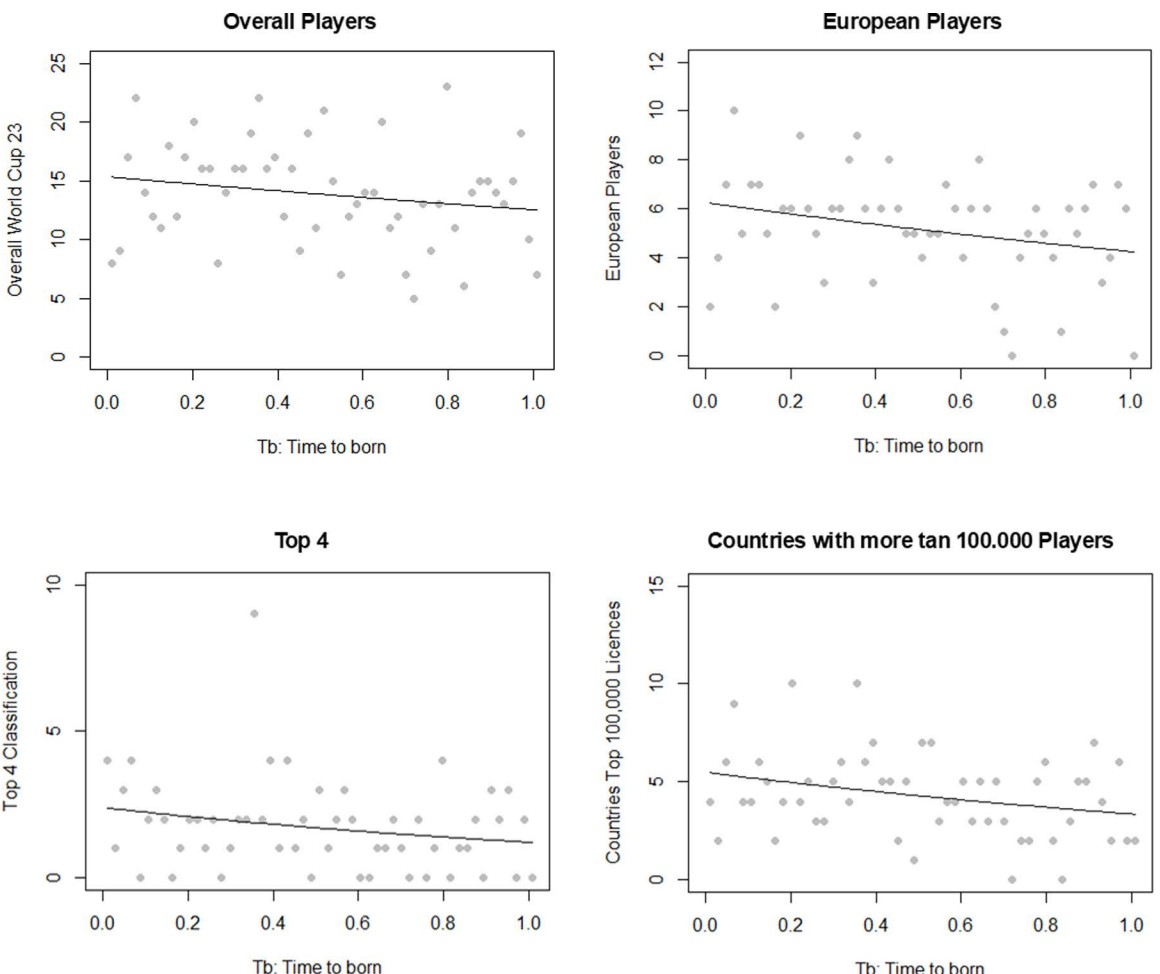

**Fig 2. Frequency of week of birth (WB) by continent and classification and Poisson Regression in Women's.**

Lastly, regarding the Pearson's correlation test, we examined the relationship between the independent variable—the federation licenses in participating countries in the Women's Football World Cup 2023—and the dependent variable—the average week of birth of the players from each country. Due to the variability in magnitude and outliers, such as the large disparity in licenses between the USA and subsequent countries, we opted for logarithmic transformation of the variables for analysis. The results are summarized in Table 3, showcasing the average week of birth of players by country and Federation Licenses.

Additionally, Table 4 exhibits the birth date distributions according to their quartile or semester of birth by demarcation, revealing a higher presence of players born in the first semester across goalkeepers, defenders, and, slightly, across midfielders. In contrast, forwards were slightly more frequently born in the second half of the year.

Fig 3 provides the frequency of week of birth (WB) for goalkeepers, defenders, midfielders, and forwards, respectively, along with Poisson Regression analyses.

Lastly, Table 5 offers a detailed Poisson regression analysis of RAE by demarcation for all players in the Women's Football World Cup 2023. This analysis did not uncover a significant RAE across any demarcation in the overall sample. However, a specific analysis of European players' demarcations revealed the presence of RAE among goalkeepers and defenders, while midfielders and forwards did not exhibit a significant RAE.

**Table 2. Poisson regression analysis of RAE by frequency for all players by Women's Football World Cup 2023.**

| Overall Players (n = 735) | $W_B$ | $26 \pm 15$ |
|---|---|---|
| | $t_B$ | $0.49 \pm 0.28$ |
| | $b_0$ | 2.73 |
| | $b_1$ | $-0.20$ |
| | $ID = \exp(b_0)/\exp(b_0 + b_1)$ | 1.22 |
| | $D^2$ (McFadden) | 0.04 |
| | *P* value | 0.11 |
| African Players (n = 297) | $W_B$ | $27 \pm 15$ |
| | $t_B$ | $0.51 \pm 0.28$ |
| | $b_0$ | 0.59 |
| | $b_1$ | $-0.08$ |
| | $ID = \exp(b_0)/\exp(b_0 + b_1)$ | 1.08 |
| | $D^2$ (McFadden) | 0.00 |
| | *P* value | 0.82 |
| American Players (n = 207) | $W_B$ | $27 \pm 15$ |
| | $t_B$ | $0.51 \pm 0.28$ |
| | $b_0$ | 0.59 |
| | $b_1$ | $-0.08$ |
| | $ID = \exp(b_0)/\exp(b_0 + b_1)$ | 1.08 |
| | $D^2$ (McFadden) | 0.00 |
| | *P* value | 0.82 |
| Asian Players (n = 115) | $W_B$ | $26 \pm 16$ |
| | $t_B$ | $0.49 \pm 0.30$ |
| | $b_0$ | 0.86 |
| | $b_1$ | $-0.17$ |
| | $ID = \exp(b_0)/\exp(b_0 + b_1)$ | 1.19 |
| | $D^2$ (McFadden) | 0.01 |
| | *P* value | 0.58 |
| European Players (n = 275) | $W_B$ | $24 \pm 15$ |
| | $t_B$ | $0.45 \pm 0.28$ |
| | $b_0$ | 1.84 |
| | $b_1$ | 0.38 |
| | $ID = \exp(b_0)/\exp(b_0 + b_1)$ | 1.47 |
| | $D^2$ (McFadden) | 0.06 |
| | *P* value | <0.05 |
| Oceanian Players (n = 46) | $W_B$ | $25 \pm 14$ |
| | $t_B$ | $0.47 \pm 0.26$ |
| | $b_0$ | 0.86 |
| | $b_1$ | $-0.17$ |
| | $ID = \exp(b_0)/\exp(b_0 + b_1)$ | 1.19 |
| | $D^2$ (McFadden) | 0.01 |
| | *P* value | 0.58 |
| Top 4 (n = 92) | $W_B$ | $24 \pm 15$ |
| | $t_B$ | $0.45 \pm 0.28$ |
| | $b_0$ | 0.88 |
| | $b_1$ | $-0.67$ |
| | $ID = \exp(b_0)/\exp(b_0 + b_1)$ | 1.96 |
| | $D^2$ (McFadden) | 0.05 |
| | *P* value | <0.05 |

*(Continued)*

**Table 2.** (Continued)

| Federation Licenses > 100,000 (n = 229) | $W_B$ | 25 ± 15 |
|---|---|---|
| | $t_B$ | 0.47 ± 0.28 |
| | $b_0$ | 1.70 |
| | $b_1$ | −0.49 |
| | ID = $\exp(b_0)/\exp(b_0 + b_1)$ | 1.63 |
| | $D^2$ (McFadden) | 0.07 |
| | *P* value | <0.05 |

$W_B$: week of birth; $t_B$: time of birth; ID: index of discrimination.

Our Poisson regression analysis reveals that there is no significant RAE in any of the demarcations in overall sample. On the other hand, the analysis of European player demarcations reveals the existence of RAE for Goalkeepers and Defenders, while there is no RAE for midfielders and forwards.

## Discussion

The analysis of the RAE within the Women's Football World Cup 2023 reveals a significant trend of birth date distributions among players, with a predominant presence of players born in the first semester across the overall player sample. This observation aligns with the established understanding of RAE, where athletes born closer to the cutoff date of sports age groups tend to have developmental and selection advantages over those born later in the year [31]. Specifically, the findings from the Women's Football World Cup 2023 showcase regional variations in RAE's manifestation, with African, American, Asian, European, and Oceanian players demonstrating differing patterns of semester birth prevalence.

In the context of African players, the stronger presence of first-semester players could reflect a regional sports development infrastructure and coordination that inadvertently amplifies RAE's impact [32]. This observation suggests that talent identification and selection processes in these countries may be influenced by physical maturity rather than skill alone, a factor that has been widely discussed in the literature as contributing to RAE [33].

Contrastingly, American players exhibited a higher frequency of second-semester players, with notable exceptions in the United States and Canada, where first-semester players were more prevalent. This could indicate an approach closer to talent development in these countries, potentially mitigating RAE through diverse selection and training practices. It is interesting noting the differences in developmental pathways of youth female soccer players. For example, in the United States, players typically do not turn professional until they are 21-23 years of age via draft system (i.e., they are adults [34]). However, many players in England engaged in senior/professional female soccer at much younger ages (17-18 years of age [35]). The persistence of RAE countries with predominance of football underscores the challenge of completely eliminating this bias from competitive sports [32].

The trend towards first-semester births in Asian, European, and Oceanian countries further corroborates the global prevalence of RAE across different sports disciplines [18,32]. Specifically, the European pattern of RAE, with the majority of countries showcasing a predominance of first-semester births, reflects established sports systems where early physical development may still be viewed as advantageous, despite increasing awareness of RAE's long-term implications and the systematic disadvantages for those born at the end of the year [33].

**Table 3. Average week of birth of players by country and Federation Licenses in participating countries in Women's Football World Cup 2023.**

| Country | Classification | Average week of birth | Federation Licenses |
|---|---|---|---|
| Spain | 1 | 25.96 | 84,658 |
| England | 2 | 22.83 | 200,967 |
| Sweden | 3 | 23.39 | 139,197 |
| Australia | 4 | 23.78 | 120,000 |
| Netherlands | 8 | 26.65 | 134,993 |
| Japan | 8 | 25.09 | 50,713 |
| Colombia | 8 | 27.38 | 26577 |
| France | 8 | 27.61 | 166,690 |
| Switzerland | 16 | 26.52 | 31,886 |
| South Africa | 16 | 27.96 | 46,319 |
| Norway | 16 | 27.35 | 94,500 |
| USA | 16 | 27.13 | 1,720,000 |
| Nigeria | 16 | 31.82 | 2,140 |
| Jamaica | 16 | 28.22 | 4,400 |
| Denmark | 16 | 23.43 | 71,588 |
| Morocco | 16 | 23.43 | 13,460 |
| New Zealand | 32 | 26 | 21,363 |
| Philippines | 32 | 25.96 | 250 |
| Canada | 32 | 23.96 | 203,586 |
| Ireland | 32 | 21.96 | 35,583 |
| Zambia | 32 | 24.36 | 4,860 |
| Costa Rica | 32 | 25.87 | 5,635 |
| China | 32 | 22.39 | 18,127 |
| Haiti | 32 | 27.09 | 600 |
| Portugal | 32 | 28.87 | 9,802 |
| Vietnam | 32 | 30.09 | 355 |
| Brazil | 32 | 26.78 | 6,351 |
| Panama | 32 | 28.13 | 1,093 |
| Italy | 32 | 26.32 | 40,207 |
| Argentina | 32 | 29.69 | 3,780 |
| Germany | 32 | 22.52 | 186,628 |
| South Korea | 32 | 27.56 | 4,600 |

While other factors, such as socioeconomic status of the continents/countries, can lead to differences regarding the RAE. For example, regions with a higher socioeconomic level typically have better access to sports facilities, quality coaches, and development programs, which could reduce these biases. Similarly, families with greater resources can provide more support, both emotional and financial, which can help mitigate the disadvantages of the RAE. In this regard, Helsen et al. [36] found a more pronounced RAE in youth soccer leagues in areas of low socioeconomic status, where resources and opportunities are more limited. Likewise, Helsen et al. [37] demonstrated that students from areas with high socioeconomic status were more likely to participate in organized sports and receive quality training, which could reduce the impact of the RAE.

Moreover, the observation that the top four countries classified in the Women's Football World Cup 2023, as well as countries with more than 100,000 Federation Licenses, exhibited

**Table 4. Birth date distributions according to their quartile (Q) or semester (Se) of birth by demarcation.**

| | | Q1 | Q2 | Q3 | Q4 | Se 1 | Se 2 |
|---|---|---|---|---|---|---|---|
| Overall Goalkeepers (n = 94) | n | 30 | 26 | 20 | 18 | 56 | 38 |
| | % | 32 | 28 | 21 | 19 | 60 | 40 |
| Overall Defenders (n = 218) | n | 56 | 63 | 53 | 46 | 119 | 99 |
| | % | 26 | 29 | 24 | 21 | 55 | 45 |
| Overall Midfielders (n = 230) | n | 53 | 64 | 55 | 58 | 117 | 113 |
| | % | 23 | 28 | 24 | 25 | 51 | 49 |
| Overall Forwards (n = 188) | n | 50 | 42 | 34 | 62 | 92 | 96 |
| | % | 27 | 22 | 18 | 33 | 49 | 51 |

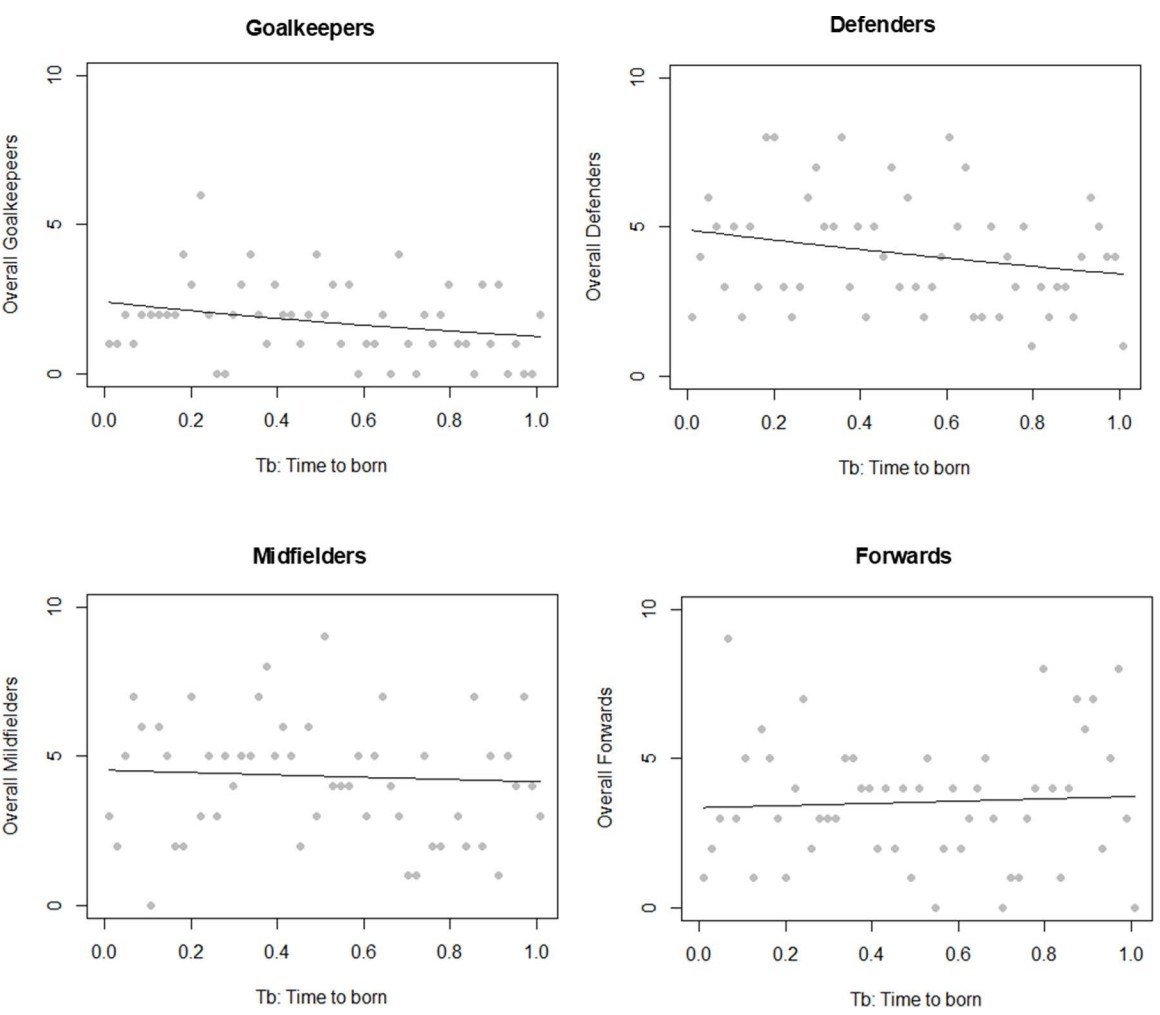

**Fig 3. Frequency of week of birth (WB) by demarcation and Poisson Regression in Women's Football World Cup 2023.**

a higher number of first-semester players, suggests that RAE may have a tangible impact on the international competitive landscape as previously observed in previous research [32]. This raises important considerations for sports governing bodies regarding the implementation of

**Table 5. Poisson regression analysis of RAE by demarcation for all players by Women's Football World Cup 2023.**

| Overall Goalkeepers (n = 94) | $W_B$ | $24 \pm 14$ |
|---|---|---|
| | $t_B$ | $0.45 \pm 0.26$ |
| | $b_0$ | 0.88 |
| | $b_1$ | -0.65 |
| | $ID = \exp(b_0)/\exp(b_0 + b_1)$ | 1.91 |
| | $D^2$ (McFadden) | 0.06 |
| | *P* value | 0.068 |
| Overall Defenders (n = 218) | $W_B$ | $25 \pm 15$ |
| | $t_B$ | $0.47 \pm 0.28$ |
| | $b_0$ | 1.59 |
| | $b_1$ | -0.36 |
| | $ID = \exp(b_0)/\exp(b_0 + b_1)$ | 1.43 |
| | $D^2$ (McFadden) | 0.05 |
| | *P* value | 0.12 |
| Overall Midfielders (n = 230) | $W_B$ | $27 \pm 15$ |
| | $t_B$ | $0.51 \pm 0.28$ |
| | $b_0$ | 1.51 |
| | $b_1$ | -0.09 |
| | $ID = \exp(b_0)/\exp(b_0 + b_1)$ | 1.09 |
| | $D^2$ (McFadden) | 0.03 |
| | *P* value | 0.69 |
| Overall Forwards (n = 188) | $W_B$ | $27 \pm 16$ |
| | $t_B$ | $0.51 \pm 0.30$ |
| | $b_0$ | 1.21 |
| | $b_1$ | 0.11 |
| | $ID = \exp(b_0)/\exp(b_0 + b_1)$ | 0.90 |
| | $D^2$ (McFadden) | 0.02 |
| | *P* value | 0.67 |
| European Goalkeepers (n = 36) | $W_B$ | $20 \pm 12$ |
| | $t_B$ | $0.37 \pm 0.22$ |
| | $b_0$ | 0.36 |
| | $b_1$ | -1.71 |
| | $ID = \exp(b_0)/\exp(b_0 + b_1)$ | 5.50 |
| | $D^2$ (McFadden) | 0.13 |
| | *P* value | <0.01 |
| European Defenders (n = 80) | $W_B$ | $23 \pm 15$ |
| | $t_B$ | $0.43 \pm 0.28$ |
| | $b_0$ | 0.82 |
| | $b_1$ | -0.88 |
| | $ID = \exp(b_0)/\exp(b_0 + b_1)$ | 2.41 |
| | $D^2$ (McFadden) | 0.12 |
| | *P* value | <0.05 |
| European Midfielders (n = 87) | $W_B$ | $26 \pm 14$ |
| | $t_B$ | $0.49 \pm 0.26$ |
| | $b_0$ | 0.57 |
| | $b_1$ | -0.15 |
| | $ID = \exp(b_0)/\exp(b_0 + b_1)$ | 1.16 |
| | $D^2$ (McFadden) | 0.03 |
| | *P* value | 0.67 |

*(Continued)*

**Table 5.** (Continued)

| European Forwards (n = 72) | $W_B$ | 29 ± 16 |
| | $t_B$ | 0.55 ± 0.30 |
| | $b_0$ | 0.04 |
| | $b_1$ | 0.50 |
| | $ID = \exp(b_0)/\exp(b_0 + b_1)$ | 0.61 |
| | $D^2$ (McFadden) | 0.03 |
| | *P* value | 0.21 |

$W_B$: week of birth; $t_B$: time of birth; ID: index of discrimination.

policies and practices that address RAE, fostering an equitable environment for talent development and selection.

The observed significance of RAE in European countries and, notably, among the top-ranked teams [top 4 and countries with more than 100,000 federation licenses] aligns with existing literature that documents a pronounced RAE in sports systems with highly competitive structures and advanced developmental programs, also observed by Cobley et al. [32]. The association between the strength of sports infrastructure, as indicated by the number of federation licenses, and the presence of RAE suggests that more developed football systems may inadvertently perpetuate RAE through their selection and development practices. According to Cobley et al. [32], coaches in developed sports environments often find themselves under significant pressure to achieve quick results in terms of performance. This pressure can lead to a challenging dilemma where the choice leans towards selecting individuals or teams who are currently more likely to secure immediate triumphs, rather than focusing on those who might promise greater success over an extended period. It is interesting to highlight that a larger number of licenses provides the coaches a larger pool of players to choose from when selecting the players for the national squad, and therefore, normally the RAE effect would be more present [38].

Conversely, the lack of a significant RAE among players from Africa, America, Asia, and Oceania could reflect differences in the sports culture, the organization of youth football, and the level of competition within these regions. For instance, less formalized or less competitive youth development systems may offer more equitable opportunities for talent to emerge, regardless of the relative age [32].

The observed distribution of birth dates according to quartile or semester of birth by demarcation [Table 4] reveals an intriguing pattern: goalkeepers and defenders show a higher prevalence of births in the first semester, while midfielders and goalkeepers only show slight differences in the semester. The demarcation-specific analysis underscores a crucial aspect of RAE's influence in women's football. Goalkeepers and defenders' roles, traditionally associated with physicality and defensive solidity, show a significant bias towards players born earlier in the selection year. This aligns with the broader literature suggesting that early-born individuals might benefit from physical and cognitive developmental advantages over their later-born peers [31,33]. In opposition, the relatively balanced distribution observed among forwards might indicate a lesser influence of physical maturity on selection in offensive positions, where skill, creativity, and agility could be valued more. The same situation is observed in European players, suggesting that, while physical and developmental advantages associated with relative age might influence player selection in defensive roles, offensive roles may prioritize attributes less directly correlated with physical maturity.

Luckily for players born in the less popular quartiles, an advantage on physical maturity and anthropometric characteristics do not translate automatically into better performance, talent development needs to be analyzed from a multifactorial holistic approach not only from

the athlete itself but also the environment that surrounds him/her [39]. There are other variables with a higher influence on the elite sportive success of a player, such as "tactical decision-making" in basketball [40] or "psychological factors" in female football [41] and Olympic athletes that will allow elite players to reach their maximum level of performance.

Nevertheless, recognizing the potential for RAE to influence player selection, especially in certain roles and regions, highlights the importance of developing strategies that mitigate its impact. These might include revising selection criteria to emphasize skill development and performance potential over immediate physical advantages and implementing coaching education programs to raise awareness of RAE, such as those suggested by Côté et al. [42].

The main limitations of this study include the potential for exploring alternative statistical methods to demonstrate the RAE effect, such as additive logistic regression [43]. Additionally, future research could incorporate other influential factors, such as the "Constituent Year Effect" [CYE], which has been described in previous literature [44]. This effect emphasizes how grouping individuals from different birth years into a single competitive age category can create disparities in developmental opportunities, warranting further consideration.

## Conclusions

The present study underlines the presence of the Relative Age Effect (RAE) within specific segments of the Women's Football World Cup 2023, particularly among European players (mainly goalkeepers and defenders), the top 4 classified teams, and nations with an extensive number of federation licenses (probably, here is the key to the problem). These findings suggest that RAE may have a notable influence on player selection processes, disproportionately benefiting those born earlier in the year. This trend was observed despite a non-significant RAE across the overall player population, indicating that the effect varies significantly across geographical regions and competitive levels.

This research deals with some limitations, mainly related to its focus on a single event and the inherent variability of the RAE across different sports and regions. While the study adds knowledge into the presence of RAE in women's football at an elite level, its applicability to broader contexts may be restricted due to these limitations.

From a practical perspective, this research could contribute to the growing call for fairer talent selection processes in sports, creating opportunities for players born throughout the year to succeed equally. The findings emphasize the need for governing bodies and coaching staff within women's football to recognize and address the implications of RAE. By implementing policies and practices that mitigate its impact, such as adjusted age-group cutoffs or diverse talent identification criteria, stakeholders can foster a more equitable and inclusive environment, ensuring players receive equal opportunities for growth and recognition in the sport.

Future studies could explore the long-term effects of RAE mitigation strategies on player development and performance outcomes in women's football. Additionally, examining the interplay between RAE and other demographic factors, such as socioeconomic status or geographical location of players, could offer deeper insights into the challenges and opportunities within talent development processes. The answers to these questions will be helpful for advancing the search of fairness and equity in sports. Furthermore, the inclusion of a control group or qualitative components could provide new perspectives to these analyses

## Author contributions

**Conceptualization:** Jairo León-Quismondo, José Bonal, Iyán Iván-Baragaño, Pablo Burillo.

**Data curation:** Benito Pérez-González.

**Formal analysis:** Benito Pérez-González.

**Investigation:** Jairo León-Quismondo, José Bonal, Iyán Iván-Baragaño.

**Supervision:** Jairo León-Quismondo, Álvaro Fernández-Luna, Pablo Burillo.

**Validation:** José Bonal, Álvaro Fernández-Luna, Pablo Burillo.

**Writing – original draft:** Iyán Iván-Baragaño, Álvaro Fernández-Luna.

**Writing – review & editing:** Benito Pérez-González, Álvaro Fernández-Luna, Pablo Burillo.

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
