## [Decision Letter · Decision Letter 0]

11 Nov 2024

PONE-D-24-47465Is the Relative Age Effect just a European problem? A Comprehensive Analysis of Birth Date Distribution and Its Impact on Player Selection at the 2023 FIFA Women's World CupPLOS ONE?

Dear Dr. Iván-Baragaño,

Thank you for submitting your manuscript to PLOS ONE. After careful consideration, we feel that it has merit but does not fully meet PLOS ONE’s publication criteria as it currently stands. Therefore, we invite you to submit a revised version of the manuscript that addresses the points raised during the review process.

We look forward to receiving your revised manuscript.

Kind regards,

Paweł Krawczyk, Ph.D.

Academic Editor

PLOS ONE

Reviewers' comments:

Reviewer's Responses to Questions

**Comments to the Author**

1. Is the manuscript technically sound, and do the data support the conclusions?

Reviewer #1: Yes

Reviewer #2: Yes

2. Has the statistical analysis been performed appropriately and rigorously?

Reviewer #1: Yes

Reviewer #2: Yes

3. Have the authors made all data underlying the findings in their manuscript fully available?

Reviewer #1: Yes

Reviewer #2: Yes

4. Is the manuscript presented in an intelligible fashion and written in standard English?

Reviewer #1: Yes

Reviewer #2: Yes

Reviewer #1: Thank you for the opportunity to review this paper. This study examines the relative age effect in female soccer players at the 2023 Women’s World Cup. Whilst this would be a good addition to the limited number of studies exploring talent identification in youth female soccer, I believe sizable changes may need to be made to the manuscript. I have outlined these below.

Introduction

• Lines 41-42: this is a nice section, but I believe it would benefit from discussing talent identification and development of female soccer in general. There are several studies that have explored various aspects of predictors and/or mediating factors such as physical skills (Datson et al., 2020; 10.1080/02640414.2019.1656323), technical (e.g., Harkness-Armstrong et al., 2020; 10.1080/24748668.2020.1820173). You could also highlight the ack of literature in this area (e.g., Curran et al., 2019; Okholm Kryger et al., 2020; 10.1080/24733938.2020.1868560) and the recent calls for more female specific literature examining talent identification in youth female soccer (e.g., Williams, Ford, Drust, 2023; ISBN 9781003375968; Emmonds et al., 2019; 10.1186/s40798-019-0224-x).

• Line 75: This is nice section, but I believe that there are several studies that may compliment this section and are not included and examine international level youth and senior female soccer layers. For example, Andrew et al. 2022 (10.3390/children9111747) explored U17, U19, and Senior Women’s soccer in the European Championships. Furthermore, Finnegan et al. 2024 explored youth national team players in the United States (10.5114/biolsport.2024.136085). Morgans et al., 2024 explored youth and senior players in Wales (10.5114/jhk%2F186563), and Gotze and Hoppe 2021 explored female soccer players in Germany (10.3389/fpsyg.2020.587023).

Methods

• Lines 78-81: The US Soccer Federation changed their cut off dates in 2017. Therefore, many players would have been in different cut off periods and therefore RAE, which is most prevalent youth ages would have been different for some players. They may have benefitted from the old system. Was this adjusted for?

• 100-101: Would it be better here to highlight which soccer federation (e.g., EUFA; CONCACAF etc.) they represent? For example, Americas would include Brazil and Canada. Two soccer nations that do not play in the same federation and also have very stark differences in socio-economics etc. that are known to influence talent identification.

• Could you clarify, is ‘Top 4’ ranked the FIFA rankings, for the WC etc? Just to be clear for the reader. It may also be worth mentioning within the introduction around RAE being linked to success, as some research has highlighted that this may be linked to success (Augste & Lames, 2011; 10.1080/02640414.2011.574719), whereas other haven’t (e.g., Andrew et al., 2022).

Results

• Well-presented tables, clear and concise.

• Lines 152 (Table 3). This is interesting, but is not worth highlighting that many of these nations have more qualified coaches simply due to the population rates as well as participation numbers?

• Figure 1 needs y and x axis labels etc.

• Figures 2-3 is very difficult to view and therefore needs to be better resolution?

Discussion

• Lines 200-202: This is a good point, but it may also be worth noting the differences in developmental pathways of youth female soccer players. For example, in the United States, players typically do not turn professional until they are 21-23 years of age via a draft system (i.e., they are adults; Ford et al., 2020; ISBN 9781003375968). However, many players in England engaged in senior/professional female soccer at much younger ages (17-18 years of age; Andrew et al., 2024; 10.1080/02640414.2024.2356434).

• Lines 212-216: Again, how many licenses per registered player may be a better discussion point? Is this possible to gather this information.

• Lines 252-254: Really good points but I think the wider implications for talent identification and development may need to be considered here.

Reviewer #2: This article explores "Relative Age Effect" (RAE)ᅳa phenomenon where athletes born earlier in the year tend to be over represented due to selection biases participate in the 2023 FIFA Women's World Cup. According athletes birth dates, the researcher aim to uncover patterns in player selection based on birth date, depending on geographical location and playing position. The study findings suggests that European athletes, especially goalkeepers and defenders, are more likely to be affected by RAE, whereas players from other regions (Africa, America, Asia, and Oceania) show no or minimal evidence of this effect. The research try to cover women's sports area, that is traditionally overlooked compared to men's sports. RAE is a interesting research area, so better selection practices could lead to more equitable representation of athletes born throughout the year, rather than skewing in favor of those born early.

The authors use a Poisson regression as method often used to predict the likelihood of an event based on an influencing factor, in this case, the birthdate of players. However, while the Poisson model is well-suited for this type of count data, it may have some limitations.

The results indicate that birth date does play a role, though it varies by region. European players, especially goalkeepers and defenders, are more likely to be born earlier in the year. This may hint at selection biases where those with early-year birthdays have a physical advantage in youth selection, which persists into professional ranks. Conversely, players from other regions don't show this pattern as strongly, suggesting their selection practices may be less influenced by birth date.

Recommendations for Improvement

1. Authors did not Analise "years of birth" (only month) and did not mention in the introduction part "constituent year effect" (CYE) witch can provide additional information for discussion.

2. The methodology is mostly appropriate, but with limitations such as the choice of statistical models, lack of controls for confounding factors, and potential selection bias. Addressing these issues could improve the robustness of the findings.

3. Elaborate on why Poisson regression was chosen and consider include additional statistical tests.

4. Comparing the players' birth dates with general population data or amateur players could provide a baseline, can help to clarify whether these trends are specific to elite football.

5. Discuss other factors (e.g., socioeconomic status) that might intersect with RAE and affect the generalization of results.

6. In the part “Further Research“ I suggest to include qualitative component or control group analysis in future research to provide additional perspectives on RAE.

Conclusions would benefit from addressing mentioned areas for a stronger, more complete analysis. This research could contribute to the growing call for fairer talent selection processes in sports, creating opportunities for players born throughout the year to succeed equally.

**Do you want your identity to be public for this peer review?** For information about this choice, including consent withdrawal, please see our Privacy Policy

Reviewer #1: **Yes: ** Matthew Andrew

Reviewer #2: **Yes: ** Full Professor Drazen Cular, PhD, University of Split, Faculty of Kinesiology, Cro Sport Talent Lab

---

## [Author Response · Author response to Decision Letter 1]

28 Dec 2024

The response is attached in the document.

---

## [Decision Letter · Decision Letter 1]

12 Jan 2025

Is the Relative Age Effect just a European problem? A Comprehensive Analysis of Birth Date Distribution and Its Impact on Player Selection at the 2023 FIFA Women's World Cup

PONE-D-24-47465R1

Dear Dr. Iván-Baragaño,

We’re pleased to inform you that your manuscript has been judged scientifically suitable for publication and will be formally accepted for publication once it meets all outstanding technical requirements.

Kind regards,

Paweł Krawczyk, Ph.D.

Academic Editor

PLOS ONE

Additional Editor Comments (optional):

Reviewers' comments:

Reviewer's Responses to Questions

**Comments to the Author**

Reviewer #1: All comments have been addressed

Reviewer #2: All comments have been addressed

2. Is the manuscript technically sound, and do the data support the conclusions?

Reviewer #1: Yes

Reviewer #2: Yes

3. Has the statistical analysis been performed appropriately and rigorously?

Reviewer #1: Yes

Reviewer #2: Yes

4. Have the authors made all data underlying the findings in their manuscript fully available?

Reviewer #1: Yes

Reviewer #2: Yes

5. Is the manuscript presented in an intelligible fashion and written in standard English?

Reviewer #1: Yes

Reviewer #2: Yes

Reviewer #1: The authors have addressed all my comments. I would like to thank them for an enjoyable read, and to the editors for providing me the opportunity to review this paper.

Reviewer #2: This study’s focus on women’s football is a valuable addition to the literature, as most RAE studies have concentrated on male athletes. It provides a nuanced understanding of how geographical and positional factors can influence RAE, potentially affecting talent identification and selection practices in sports. However, after first revision the manuscript is updated according my recommendations & instructions and this manuscript is now acceptable for publication.

**Do you want your identity to be public for this peer review?** For information about this choice, including consent withdrawal, please see our Privacy Policy

Reviewer #1: **Yes: ** Matthew Andrew

Reviewer #2: **Yes: ** prof. Dražen Čular, PhD, Cro Sport Talent Lab, University of Split, Faculty of Kinesiology

---

## [Editor Report · Acceptance letter]

PONE-D-24-47465R1

PLOS ONE

Dear Dr. Iván-Baragaño,

I'm pleased to inform you that your manuscript has been deemed suitable for publication in PLOS ONE. Congratulations! Your manuscript is now being handed over to our production team.

Kind regards,

on behalf of

Dr. Paweł Krawczyk

Academic Editor

PLOS ONE